# Automated extracellular volume fraction measurement for diagnosis and prognostication in patients with light-chain cardiac amyloidosis

In-Chang Hwang [1,2]*, Eun Ju Chun [3,4]*, Pan Ki Kim[5], Myeongju Kim[6], Jiesuck Park[1,2], Hong-Mi Choi[1,2], Yeonyee E. Yoon[1,2], Goo-Yeong Cho[1,2], Byoung Wook Choi[5,7]

1 Cardiovascular Center, Seoul National University Bundang Hospital, Seongnam, Gyeonggi, South Korea, 2 Department of Internal Medicine, Seoul National University College of Medicine, Seoul, South Korea, 3 Department of Radiology, Seoul National University Bundang Hospital, Seongnam, Gyeonggi, South Korea, 4 Department of Radiology, Seoul National University College of Medicine, Seoul, South Korea, 5 Phantomics, Inc., Seoul, South Korea, 6 Center for Artificial Intelligence in Healthcare, Seoul National University Bundang Hospital, Seongnam, Gyeonggi, South Korea, 7 Department of Radiology, Research Institute of Radiological Science, Center for Clinical Imaging Data Science, Yonsei University College of Medicine, Seoul, South Korea

☯ These authors contributed equally to this work.
* inchang.hwang@gmail.com (I-CH); humandr@snubh.org (EJC)

## Abstract

### Aims

T1 mapping on cardiac magnetic resonance (CMR) imaging is useful for diagnosis and prognostication in patients with light-chain cardiac amyloidosis (AL-CA). We conducted this study to evaluate the performance of T1 mapping parameters, derived from artificial intelligence (AI)-automated segmentation, for detection of cardiac amyloidosis (CA) in patients with left ventricular hypertrophy (LVH) and their prognostic values in patients with AL-CA.

### Methods and results

A total of 300 consecutive patients who underwent CMR for differential diagnosis of LVH were analyzed. CA was confirmed in 50 patients (39 with AL-CA and 11 with transthyretin amyloidosis), hypertrophic cardiomyopathy in 198, hypertensive heart disease in 47, and Fabry disease in 5. A semi-automated deep learning algorithm (Myomics-Q) was used for the analysis of the CMR images. The optimal cutoff extracellular volume fraction (ECV) for the differentiation of CA from other etiologies was 33.6% (diagnostic accuracy 85.6%). The automated ECV measurement showed a significant prognostic value for a composite of cardiovascular death and heart failure hospitalization in patients with AL-CA (revised Mayo stage III or IV) (adjusted hazard ratio 4.247 for ECV ≥40%, 95% confidence interval 1.215–14.851, p-value = 0.024). Incorporation of automated ECV measurement into the revised Mayo staging system resulted in better risk stratification (integrated discrimination index 27.9%, p = 0.013; categorical net reclassification index 13.8%, p = 0.007).

**Data Availability Statement:** Data used in this study cannot be made publicly available because of the strict ethical restrictions set by the IRB of Seoul

National University Bundang Hospital (https://e-irb.snubh.org). Please contact the corresponding authors (inchang.hwang@gmail.com or humandr@snubh.org) or the ethics board at SNUBH (snubhirb@gmail.com) for further inquiries regarding data availability within the scope permitted by the IRB. The code for statistical analysis utilized in the present study was released (https://github.com/Inchang-Hwang/CMR-ECV-for-AL-CA).

**Funding:** This work was supported by the Medical AI Clinic Program through the National IT Industry Promotion Agency (NIPA), funded by the Ministry of Science and ICT (MSIT) of the Republic of Korea, and by research grants from Seoul National University Bundang Hospital (Grant Nos. 02-2017-0040 and 06-2020-0130). Pan Ki Kim and Byoung Wook Choi are founders of Phantomics, Inc. (Seoul, Korea), which provided support for the software used in this study. Phantomics, Inc. also funded salaries for PKK and BWC but had no role in the study design, data collection, analysis, publication decisions, or manuscript preparation. The specific contributions of each author are detailed in the 'author contributions' section. No additional external funding was received for this study.

**Competing interests:** Pan Ki Kim and Byoung Wook Choi are founders of Phantomics, Inc. (Seoul, Korea), which provided support for the software used in this study. Phantomics, Inc. also funded salaries for PKK and BWC but had no influence on the study's design, data collection, analysis, publication decisions, or manuscript preparation. This statement does not alter the authors' adherence to PLOS ONE policies on data and material sharing. All other authors (ICH, EJC, MK, JP, HMC, YEY, and GYC) declare that they have no competing interests related to Phantomics, Inc. or any other funders or institutions, including any affiliations involving employment, consultancy, patents, products in development, or marketed products. The specific contributions of each author are detailed in the 'author contributions' section.

## Conclusions

T1 mapping on CMR imaging, derived from AI-automated segmentation, not only allows for improved diagnosis of CA from other etiologies of LVH, but also provides significant prognostic value in patients with AL-CA.

## Introduction

Differential diagnosis of left ventricular hypertrophy (LVH) is one of the most challenging issues in cardiovascular imaging [1]. The different etiologies of LVH have morphological similarities, but unique pathophysiology, treatment strategy, and prognosis [2,3]. In particular, differential diagnosis of cardiac amyloidosis (CA) is critical for effective management: light-chain cardiac amyloidosis (AL-CA) is a hematologic malignancy that requires cytotoxic chemotherapy and/or stem cell transplantation, whereas transthyretin CA (TTR-CA) is a rare infiltrative cardiomyopathy that requires specific treatments such as TTR stabilization [4].

While several combinations of clinical, electrocardiographic, and echocardiographic features have shown acceptable accuracy, cardiac magnetic resonance (CMR) remains the most important imaging modality for the differential diagnosis of LVH [4]. The presence and patterns of late gadolinium enhancement (LGE) have proven to be relevant markers for the detection of CA and hypertrophic cardiomyopathy (HCM) from hypertensive heart disease (HHD) [2,5]. In addition, novel parameters of myocardial fibrosis, including native T1 and extracellular volume fraction (ECV), have proven to be potentially useful for differential diagnosis and prognostication in patients with various etiologies of LVH, especially in those with CA [6]. Further, it is not well-established whether the T1 mapping parameters have additive prognostic value in patients with AL-CA.

In the present study, we aimed to assess the diagnostic performance of the artificial intelligence (AI)-automated segmentation T1 mapping for the detection of CA in patients with LVH. In addition, we analyzed the prognostic value of the automated ECV measurements in patients with AL-CA in comparison with current clinical staging system.

## Methods

### Study design and cohort

This study was conducted in accordance with the principles outlined in the 2013 revised Declaration of Helsinki and approved by Seoul National University Bundang Hospital Institutional Review Board (IRB No. B-2208-773-108). The requirement for informed consent was waived owing to the retrospective nature of the study and the minimal expected risk to the patients.

We retrospectively identified 300 consecutive patients (47 with HHD, 198 with HCM, 50 with CA [39 with AL-CA, and 11 with TTR-CA], and 5 with Fabry disease [FD]) (**Fig 1**) who underwent CMR for differential diagnosis of the etiology, based on the detection of LVH on echocardiography, between 2011 and 2023. Data for the study population were accessed from June 14th, 2023, to August 4th, 2023, for this research. All data were fully anonymized before access. The diagnostic criteria for HHD, HCM, AL-CA, TTR-CA, and FD are described below.

**Hypertensive heart disease.** Patients with a history of hypertension who met the diagnostic criteria for LVH on echocardiography (LV mass index [LVMI] >115 g/m$^2$ for men, and >95 g/m$^2$ for women), without other potential causes of LVH [7].

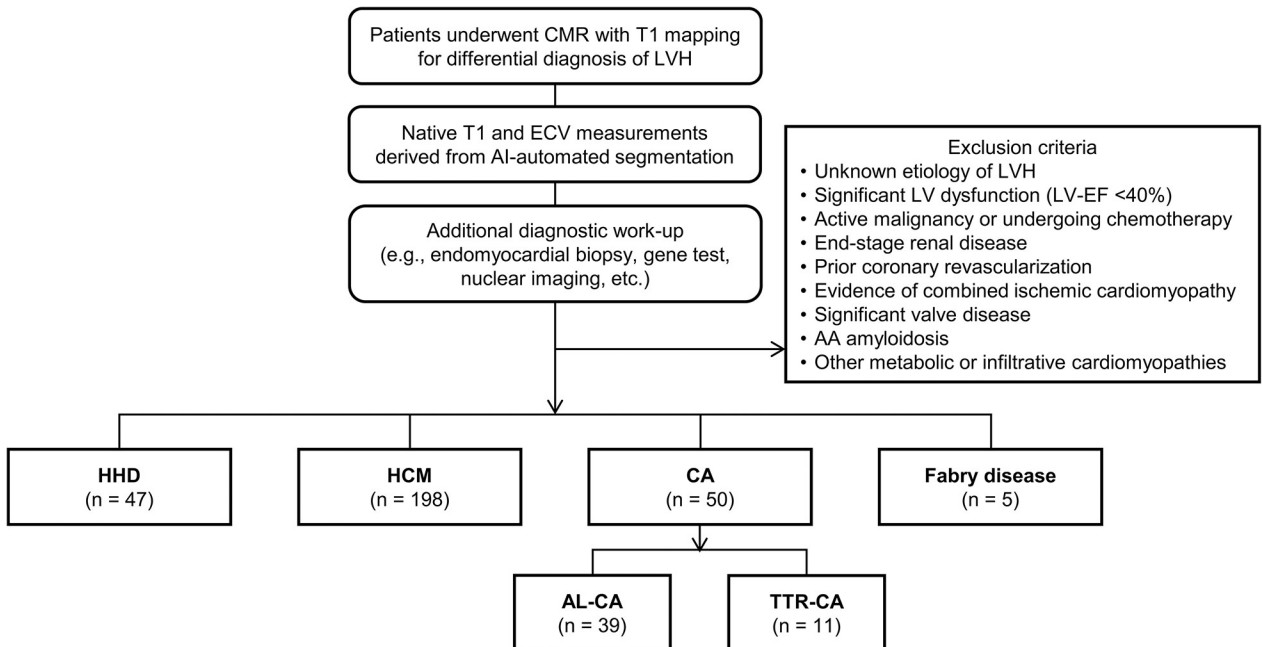

**Fig 1. Flowchart of patient selection.** Abbreviations: CMR, cardiac magnetic resonance; LVH, left ventricular hypertrophy; AI, artificial intelligence; ECV, extracellular volume fraction; HHD, hypertensive heart disease; HCM, hypertrophic cardiomyopathy; CA, cardiac amyloidosis; AL-CA, AL cardiac amyloidosis; TTR-CA, transthyretin cardiac amyloidosis.

**Hypertrophic cardiomyopathy.** Patients who met the diagnostic criteria of HCM (LVWTmax ≥15 mm on echocardiography, in the absence of abnormal loading conditions that could sufficiently explain the LVH) were included [8,9]. Definite evidence of HCM on CMR or identification of a typical gene mutation was required for a precise diagnosis.

**Cardiac amyloidosis.** The types of CA were determined by evaluating serum and urine electrophoresis, serum free light chains, CMR imaging, and technetium-99m scintigraphy, and were confirmed based on the findings of myocardial biopsies [5]. AL-CA was defined as biopsy-proven AL amyloidosis and/or monoclonal gammopathy, accompanied by the following findings: (i) increased LV wall thickness without dilated LV (average LV wall thickness ≥12 mm) in the presence of low voltage QRS (amplitude <0.5 mV in the limb leads or <1.0 mV in the anterior leads); or (ii) typical findings on CMR (patchy, subendocardial circumferential, or diffuse fuzzy LGE of the LV). Patients who showed positive or equivocal TTR immunohistochemical staining on myocardial biopsy underwent direct screening analysis for detection of *TTR* mutations [10].

**Fabry disease.** FD was diagnosed based on the assessment of α-galactosidase A activity in the serum (for male patients) and the detection of *GLA* mutation (for both male and female patients) [11].

**Exclusion criteria.** Patients with unknown etiology of LVH even after performing all available diagnostic tests were excluded. In addition, patients were excluded if they presented with any of the following: (1) significant LV dysfunction (LV ejection fraction <40%), (2) active malignancy (other than AL-CA) or undergoing chemotherapy, (3) end-stage renal disease, (4) prior coronary revascularization, (5) evidence of combined ischemic cardiomyopathy, (6) significant valve disease, (7) AA amyloidosis, or (8) other metabolic or infiltrative cardiomyopathies.

## Clinical data acquisition

Clinical and anthropometric, laboratory, and echocardiographic measurements were obtained at the time of CMR imaging. All echocardiographic images were obtained using a standard ultrasound device with a 2.5-MHz probe, in accordance with the guidelines of the European Association of Cardiovascular Imaging [7].

## CMR imaging acquisition and assessment of myocardial fibrosis

Detailed procedure used for CMR imaging is provided in the (**S1 File**). CMR imaging was performed according to standard protocols using a 3.0-T imager (Ingenia CX, Philips Healthcare, Best, Netherlands) with a 16-channel phased array coil. Cine imaging was obtained using a segmented steady-state free-precession sequence. LGE images were obtained using a phase-sensitive inversion recovery sequence 10 min after the injection of 0.2 mmol/kg of gadobutrol (Gadovist, Bayer Schering Pharma, Berlin, Germany). Pre- and post-contrast (15 minutes) T1 mapping was performed using a mid-ventricular short-axis section at the level of papillary muscles, and the images were acquired using the modified Look-Locker inversion-recovery (MOLLI) sequence (the "3-3-5" standard protocol) [6]. The region of interest was delineated on the entire LV myocardium at the level of papillary muscles [12,13].

T1 maps were generated using MR workstation with in-line motion correction following image acquisition. The recovery rate of T1 relaxation was measured in a mid-ventricular short-axis slice [13,14]. ECV was calculated as ECV = $(1 - \text{hematocrit}) \times [\triangle R1_{\text{myocardium}}] / [\triangle R1_{\text{blood}}]$, with hematocrit measured at the time of CMR imaging [6]. LGE extent was categorized into focal (localized distribution), multifocal (distribution across several locations), and diffuse (involving at least three contiguous myocardial segments). Additionally, the distribution patterns were classified into six distinct types and analyzed as follows; patchy segmental, transmural, mid-layer, RV insertion point, subendocardial ring, and nonspecific. All measurements were performed by a single radiologist (E-J.C.).

## AI-based assessment of native T1 and ECV

Our deep learning (DL) algorithm was used for automated analysis of T1 maps in accordance with the method described in the previous study, in which 2D U-Net for myocardial segmentation incorporated into Myomics-T1 software ver. 1.0.0 (Phantomics Inc.) [13]. Details of the architecture of the DL model are presented in the (**S1 File**) and **S1 Fig**. Additional relevant information can be found in previous publications [13,14]. Representative figures of the native T1 and ECV measurements derived from AI-automated segmentation for various etiologies of LVH are shown in **Fig 2**. The correlation and agreement analyses conducted using 30 randomly selected patients are summarized in **S2 Fig**.

## Study outcomes

The diagnostic outcome was the area under the receiver operating characteristic curve (AUC) for differentiating CA from other etiologies of LVH. The clinical outcome for prognosis of AL-CA was a composite of cardiovascular death and hospitalization for heart failure, assessed through regular outpatient visits, telephone interviews and chart reviews.

## Statistical analysis

Categorical variables are presented as frequencies and percentages, and continuous variables as means ± standard deviations. Group comparisons were performed with Mann-Whitney U test and Kruskal-Wallis test for continuous variables, and the $\chi 2$ test or Fisher's exact test for

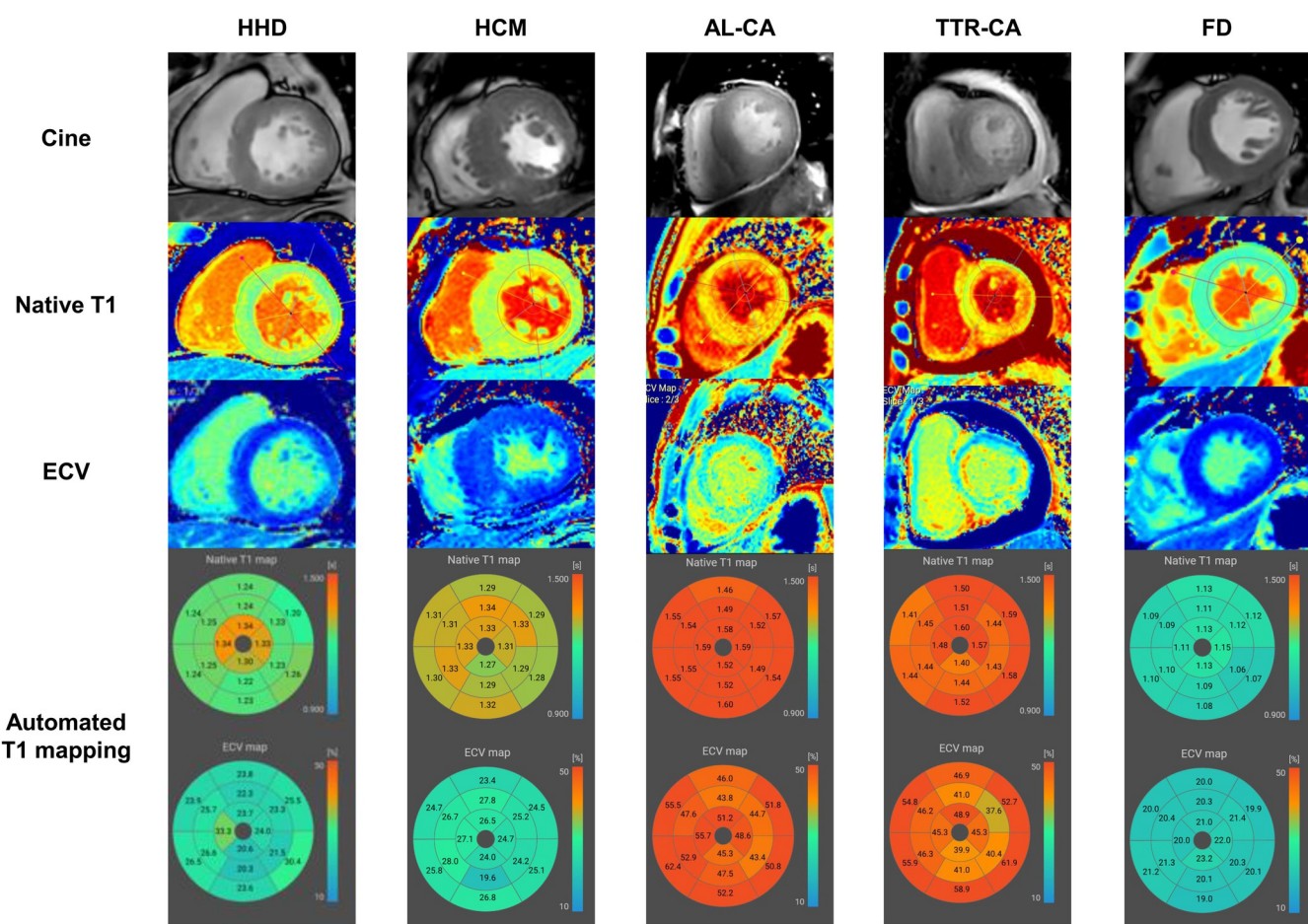

**Fig 2. Representative figures of AI-automated segmentation T1 mapping.** Typical native T1 and ECV measurements obtained through an AI-automated segmentation T1 mapping for various etiologies of LVH: **(A)** HHD, **(B)** HCM, **(C)** AL-CA, **(D)** TTR-CA, and **(E)** FD. Abbreviations: FD, Fabry disease; others as in Fig 1.

categorical variables. The AUC was used to measure the performance of the DL algorithm in the diagnosis of CA. Kaplan–Meier method and Cox proportional hazard regression model were used for survival analysis. Multivariate Cox proportional hazards regression with backward selection method was performed using univariable markers with p-values <0.100. All statistical analyses were performed using R statistical software version 3.6.3 (The R Foundation, Vienna, Austria), and p-value <0.05 was considered statistically significant.

## Results

### Baseline characteristics

Baseline characteristics of the study population are summarized in **Table 1**. The mean LVMI was 118.7±33.7 g/m$^2$ in patients with HHD, 134.9±32.7 g/m$^2$ in those with HCM, 134.0±36.1 g/m$^2$ in those with AL-CA, 143.0±54.0 g/m$^2$ in those with TTR-CA, and 162.2±80.2 g/m$^2$ in those with FD. LV-EDV measured by CMR was smallest in patients with AL-CA (74.0±24.9 mL), followed by those with TTR-CA (81.2±24.4 mL), and largest in patients with Fabry disease (107.7±57.8 mL). LV-EF measured by CMR was comparable among patients with HHD, HCM, and Fabry disease but was significantly lower in those with AL-CA (59.0±11.0%). LGE

**Table 1. Baseline characteristics.**

| | HHD (n = 47) | HCM (n = 198) | CA (n = 50) | | Fabry disease (n = 5) | p-value |
|---|---|---|---|---|---|---|
| | | | AL-CA (n = 39) | TTR-CA (n = 11) | | |
| **Age, years** | 56.4±16.3 | 57.6±13.8 | 69.8±10.4 | 78.7±5.3 | 45.9±3.3 | <0.001 |
| **Male** | 30 (63.8%) | 143 (72.2%) | 24 (61.5%) | 3 (27.3%) | 4 (80.0%) | 0.022 |
| **Body-mass index, kg/m²** | 24.8±3.6 | 25.7±3.9 | 24.2±2.7 | 21.9±9.2 | 22.5±2.2 | 0.028 |
| **Systolic BP, mmHg** | 136.8±22.8 | 130.3±15.9 | 115.4±20.4 | 128.5±23.3 | 140.8±12.5 | <0.001 |
| **Diastolic BP, mmHg** | 81.9±15.4 | 76.2±12.2 | 70.4±12.7 | 68.5±4.9 | 79.8±7.5 | 0.001 |
| **Hypertension** | 47 (100.0%) | 66 (33.3%) | 13 (33.3%) | 1 (9.1%) | 0 (0.0%) | 0.129 |
| **Diabetes mellitus** | 13 (27.7%) | 49 (24.7%) | 10 (25.6%) | 1 (9.1%) | 0 (0.0%) | 0.641 |
| **Hemoglobin, g/dL** | 13.7±2.3 | 14.4±1.8 | 11.8±1.6 | 10.7 ± 0.6 | 13.2 ± 1.0 | <0.001 |
| **Hematocrit, %** | 40.8±6.3 | 42.9±5.1 | 35.7±4.6 | 34.0±0.2 | 40.2±3.1 | <0.001 |
| **Serum creatinine, mg/dL** | 1.31±1.81 | 0.87±0.22 | 1.02±0.43 | 0.99±0.52 | 0.97±0.19 | 0.015 |
| **GFR, mL/min/1.73m²** | 80.9±27.5 | 90.1±16.7 | 73.3±25.6 | 63.0±39.6 | 89.0±12.6 | <0.001 |
| **CKMB, ng/mL** | 6.2±10.5 | 11.7±49.1 | 4.6±3.2 | 2.9±2.8 | 5.7±1.8 | 0.901 |
| **Troponin I, ng/mL** | 1.05±3.22 | 1.65±5.40 | 0.48±0.73 | 0.04±0.00 | 0.34±0.59 | 0.664 |
| **NT-proBNP, pg/mL** | 1391.0±3055.5 | 1832.2±2383.8 | 8023.6±6639.1 | 1327.9±1119.5 | 1670.2±2778.8 | <0.001 |
| **Echocardiographic parameters** | | | | | | |
| **LV-EDV, mL** | 83.6±27.9 | 77.4±25.9 | 67.6±19.1 | 64.5±12.0 | 95.8±19.5 | 0.021 |
| **LV-ESV, mL** | 34.3±18.3 | 28.2±13.1 | 30.3±11.8 | 31.0±22.6 | 29.6±6.7 | 0.125 |
| **LV-EF, %** | 60.3±9.5 | 63.7±7.2 | 55.6±8.8 | 54.4±26.6 | 69.2±1.9 | <0.001 |
| **LVMI, g/m²** | 118.7±33.7 | 134.9±32.7 | 134.0±36.1 | 143.0±54.0 | 162.2±80.2 | 0.017 |
| **LAVI, mL/m²** | 42.6±16.4 | 50.7±19.5 | 54.8±14.6 | 63.0±22.5 | 44.0±11.1 | 0.020 |
| **E/e' ratio** | 13.5±6.8 | 14.6±6.8 | 26.0±8.9 | 28.5±8.9 | 17.6±2.7 | <0.001 |
| **TR Vmax, m/sec** | 2.5±0.4 | 2.4±0.3 | 4.2±8.1 | 3.3±0.1 | 2.5±0.5 | 0.034 |
| **LV-GLS, %** | 13.9±2.9 | 11.9±3.8 | 9.5±3.4 | 14.8±4.0 | 11.5±5.0 | <0.001 |
| **CMR measurements** | | | | | | |
| **LV-EDV by CMR, mL** | 85.7±23.4 | 87.8±28.2 | 74.0±24.9 | 81.2±24.4 | 107.7±57.8 | 0.022 |
| **LV-ESV by CMR, mL** | 30.5±18.1 | 29.7±16.8 | 31.1±16.2 | 31.0±11.5 | 38.6±26.8 | 0.816 |
| **LV-EF by CMR, %** | 66.2±12.7 | 67.3±10.6 | 59.0±11.0 | 61.2±11.5 | 65.2±5.9 | 0.001 |
| **Presence of LGE** | 19 (40.4%) | 192 (97.0%) | 38 (97.4%) | 11 (100.0%) | 4 (80.0%) | <0.001 |
| **Extent of LGE** | | | | | | <0.001 |
| • **Focal** | 14 (29.8%) | 77 (38.9%) | 2 (5.1%) | 0 (0.0%) | 2 (40.0%) | |
| • **Multifocal** | 3 (6.4%) | 104 (52.5%) | 1 (2.6%) | 1 (9.1%) | 2 (40.0%) | |
| • **Diffuse** | 2 (4.3%) | 11 (5.6%) | 35 (89.7%) | 10 (90.9%) | 0 (0.0%) | |
| **Distribution pattern of LGE** | | | | | | |
| • **Patchy segmental** | 8 (17.0%) | 122 (61.6%) | 2 (5.1%) | 0 (0.0%) | 3 (60.0%) | <0.001 |
| • **Transmural** | 0 (0.0%) | 7 (3.5%) | 6 (15.4%) | 2 (18.2%) | 0 (0.0%) | 0.002 |
| • **Mid-layer** | 0 (0.0%) | 2 (1.0%) | 0 (0.0%) | 0 (0.0%) | 0 (0.0%) | 0.904 |
| • **RV insertion point** | 9 (19.1%) | 70 (35.4%) | 1 (2.6%) | 0 (0.0%) | 0 (0.0%) | <0.001 |
| • **Subendocardial ring** | 1 (2.1%) | 7 (3.5%) | 35 (89.7%) | 11 (100.0%) | 1 (20.0%) | <0.001 |
| • **Nonspecific** | 1 (2.1%) | 0 (0.0%) | 0 (0.0%) | 0 (0.0%) | 0 (0.0%) | 0.249 |
| **Native T1, msec** | 1291.4±38.9 | 1319.4±56.9 | 1444.6±83.4 | 1418.3±72.8 | 1143.9±80.3 | <0.001 |
| **Post T1, msec** | 486.1±47.2 | 451.2±71.9 | 397.3±79.9 | 365.3±73.9 | 496.9±57.1 | <0.001 |
| **Post-contrast T2, msec** | 49.6±4.0 | 49.5±3.1 | 61.6±11.6 | 52.3±2.8 | 46.8±4.8 | <0.001 |
| **ECV, %** | 26.1±3.3 | 29.5±5.9 | 43.6±8.1 | 40.0±4.3 | 23.2±3.0 | <0.001 |

Values are presented as the mean±standard deviation or as a number (percentage).

Abbreviations: BP, blood pressure; GFR, glomerular filtration rate; CKMB, creatine kinase-myocardial band; NT-proBNP, N-terminal pro-B natriuretic peptide; LV, left ventricular; EDV, end-diastolic volume; ESV, end-systolic volume; EF, ejection fraction; LAVI, left atrial volume index; LVMI, left ventricular mass index; TR, tricuspid regurgitation; LV-GLS, left ventricular global longitudinal strain; ECV, extracellular volume fraction; CMR, cardiac magnetic resonance.

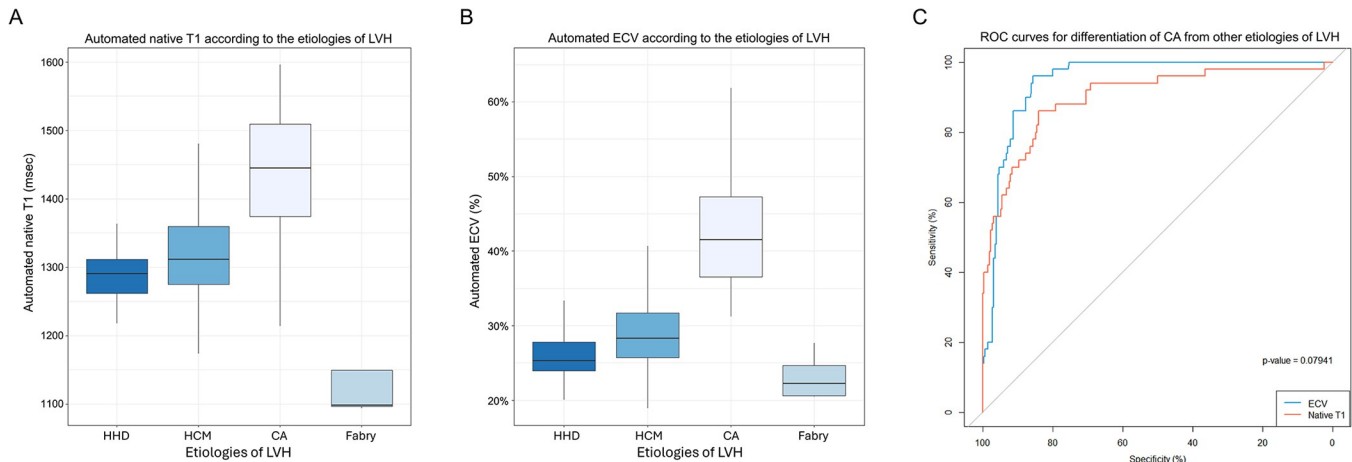

**Fig 3. Automated T1 mapping parameters and differentiation of LVH etiologies.** Automated native T1 (A) and ECV (B) measurements are shown across various etiologies of LVH. (C) ROC curves for the differentiation of CA from other LVH etiologies using native T1 and ECV. Abbreviations: ROC, receiver operating characteristic; others as in Fig 1.

was observed in the majority of patients with HCM (97.0%), AL-CA (97.4%), TTR-CA (100.0%), and Fabry disease (80.0%). In contrast, only 40% of patients with HHD showed LGE, which primarily presented as focal, patchy enhancement. Patients with AL-CA and TTR-CA typically exhibited diffuse LGE with a subendocardial ring-like pattern, while those with HCM demonstrated focal or multifocal LGE with patchy, segmental enhancement. Patients with CA had the highest native T1 values (1444.6±83.4 msec in AL-CA, and 1418.3 ±72.8 msec in TTR-CA), followed by patients with HCM (1319.4±56.9 msec), and those with HHD (1291.4±38.9 msec), and patients with FD (1143.9±80.3 msec) (**Fig 3A**). The ECV values showed similar trends: the highest in patients with AL-CA (43.6±8.1%), followed by those with TTR-CA (40.0±4.3%), HCM (29.5±5.9%), and HHD (26.1±3.3%), and FD (23.2±3.0%) (**Fig 3B**).

## Accuracy of automated native T1 and ECV measurements

The automated and manual native T1 and ECV measurements were compared on a per-patient basis (**S2 Fig**). Automated native T1 showed a strong correlation and good agreement with the reference native T1 (r = 0.995, p<0.001; bias 2.3 msec, 95% limits of agreement [LoA] -15.0 to 19.6 msec). In addition, automated ECV was strongly correlated with the reference ECV (r = 0.993, p<0.001) with a good agreement (bias 0.2%, 95% LoA -2.2% to 2.6%).

## Differentiation of CA from other etiologies of LVH using native T1 and ECV

The diagnostic performance of the automated native T1 and ECV measurements was assessed in terms of differentiating CA from other etiologies of LVH. The AUC values for the diagnosis of CA were 0.899 (95% CI 0.847–0.952; p<0.001) for native T1 and 0.946 (95% CI 0.922–0.971; p<0.001) for ECV (**Fig 3C**). The optimal native T1 and ECV cutoff values were 1364 msec and 33.6%, respectively (accuracy, 84.0% and 85.6%; sensitivity, 84.5% and 87.5%; and specificity, 86.8% and 96.2%, respectively).

## Prognostication with native T1 and ECV for AL-CA

Thirty-nine patients in the study population were diagnosed with AL-CA: 14 (35.9%) were at revised Mayo stage III and 25 (64.1%) were at stage IV. During a mean 22 months of follow-

up, 17 patients died of cardiovascular causes and 20 patients were hospitalized for heart failure. The AUC values for the prediction of study outcome using native T1 and ECV were 0.658 (95% CI 0.462–0.855; p = 0.139) and 0.888 (95% CI 0.764–1.000; p<0.001), respectively (p-for-comparison = 0.05; **S3 Fig**). The optimal cutoff value of ECV for study outcome was 40.0% (accuracy, 92.9%; sensitivity, 79.0%; and specificity, 70.8%).

Survival curves of patients with AL-CA are depicted in **Fig 4**, according to the difference in the serum free light chain (dFLC) (**Fig 4A**), NT-proBNP (**Fig 4B**), revised Mayo stage

### A. delta FLC

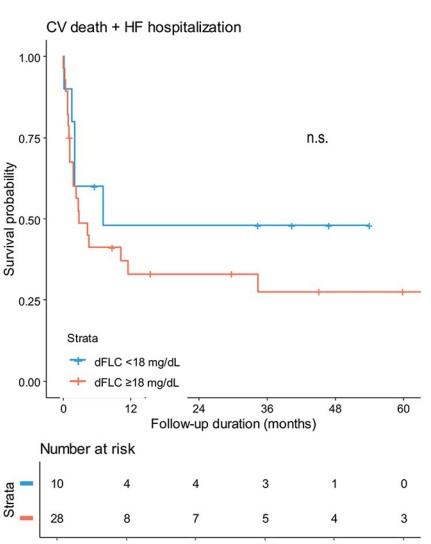

### B. NT-proBNP

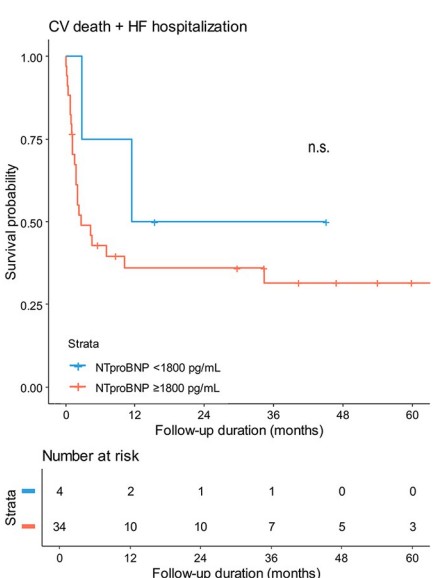

### C. Revised Mayo stage

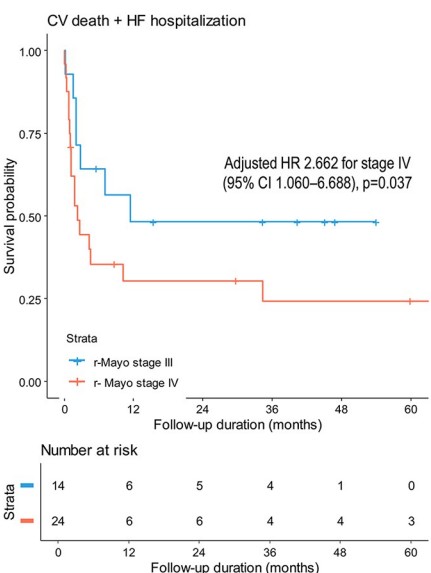

### D. Automated ECV

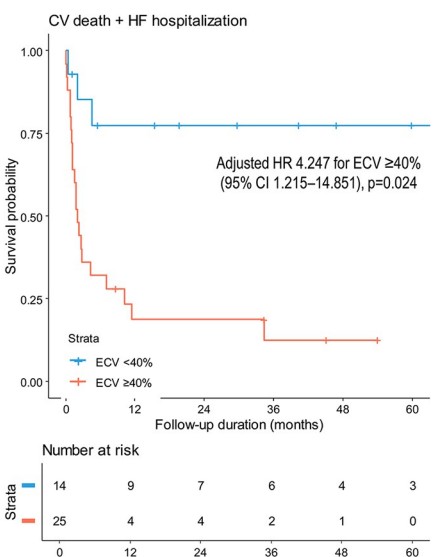

**Fig 4. Event-free survival of patients with AL-CA.** Clinical outcomes were compared between the subgroups divided by **(A)** dFLC, **(B)** NT-proBNP, **(C)** revised Mayo stage, and **(D)** automated ECV. Abbreviations: dFLC, difference in the serum levels of free light chain; NT-proBNP, N-terminal pro-B natriuretic peptide; n.s., not significant; others as in Fig 1.

Table 2. Multivariable predictors of composite outcome among patients with AL cardiac amyloidosis.

| | Multivariable analysis using continuous variables | | Multivariable analysis using categorical variables | | Multivariable analysis using revised Mayo stages combined with ECV | |
|---|---|---|---|---|---|---|
| | Adjusted HR (95% CI) | P-value | Adjusted HR (95% CI) | P-value | Adjusted HR (95% CI) | P-value |
| Systolic BP (per +1 mmHg) | 0.963 (0.935–0.992) | 0.014 | | | | |
| Systolic BP <110 mmHg | | | 2.575 (1.074–6.173) | 0.034 | 1.478 (0.493–4.435) | 0.485 |
| NT-proBNP (per +1 pg/mL) | 1.000 (1.000–1.000) | 0.847 | | | | |
| Troponin T ≥0.025 ng/mL | | | 1.039 (0.372–2.904) | 0.942 | | |
| LVGLS (per +1%) | 0.850 (0.703–1.027) | 0.092 | | | | |
| LVGLS <8% | | | 1.888 (0.787–4.525) | 0.154 | | |
| LVGLS <10% | | | | | 2.116 (0.806–5.559) | 0.128 |
| Revised Mayo stage | | | | | 2.662 (1.060–6.688) | 0.037 |
| ECV (per +1%) | 1.080 (1.011–1.153) | 0.023 | | | | |
| ECV ≥40% | | | 4.247 (1.215–14.851) | 0.024 | 6.324 (1.794–22.297) | 0.004 |

Abbreviations: HR, hazard ratio; CI, confidence interval; others as in Table 1.

(**Fig 4C**), and automated ECV (**Fig 4D**). The automated ECV measurements showed a significant discrimination of prognosis (adjusted HR 4.247 for ECV ≥40%, 95% CI 1.215–14.851, p = 0.024), whereas other conventional risk parameters, such as the dFLC, NT-proBNP, cardiac troponin, and LV global longitudinal strain, did not show significant prognostic value (**Tables 2** and S1). The prognostic value of the automated ECV remained significant even after adjusting for the revised Mayo stage (adjusted HR 6.324 for ECV ≥40%, 95% CI 1.794–22.297, p = 0.004). The extent and distribution pattern of LGE were not associated with the study outcomes due to the predominantly similar LGE patterns observed in patients with AL-CA: specifically, 35 patients (89.7%) with AL-CA exhibited diffuse LGE with a characteristic subendocardial ring-type distribution (**Tables 1** and S1).

The prognostic value of the automated ECV in patients with AL-CA was further assessed using the following subgroups: group 1 (revised Mayo stage III and ECV <40%), group 2 (revised Mayo stage IV and ECV <40%), group 3 (revised Mayo stage III and ECV ≥40%), and group 4 (revised Mayo stage IV and ECV ≥40%). The combination of revised Mayo stage and automated ECV showed excellent risk stratification, with the worse prognosis in group 4 and the best in group 1 (**Fig 5**). The AUC of the combination of revised Mayo stage and automated ECV was significantly higher than that of the Mayo staging system alone (AUC 0.860 vs. 0.588; p = 0.006). In addition, net reclassification improvement (NRI) and integrated discrimination improvement (IDI) analyses showed that the combination of ECV and the revised Mayo staging system could provide significantly better risk stratification than the revised Mayo staging system alone (IDI 0.279, 95% CI 0.025–0.499, p = 0.013; NRI 0.138, 95% CI 0.000–0.597, p = 0.007; **Table 3**).

## Prognostic value of ECV in patients with TTR

The prognostic value of the automated ECV was further assessed in patients with TTR-CA. Among the 11 patients with TTR-CA included in the present study (3 with hereditary TTR-CA [genetic; hATTR] and 8 with wild-type TTR-CA [senile; wtATTR]), 1 patient died of cardiovascular cause and 6 patients experienced hospitalization for heart failure. The AUC values for the study outcome using the automated native T1 and ECV were 0.600 (95% CI 0.226–0.974; p = 0.584) and 0.700 (95% CI 0.376–1.000; p = 0.273), respectively. These patients with

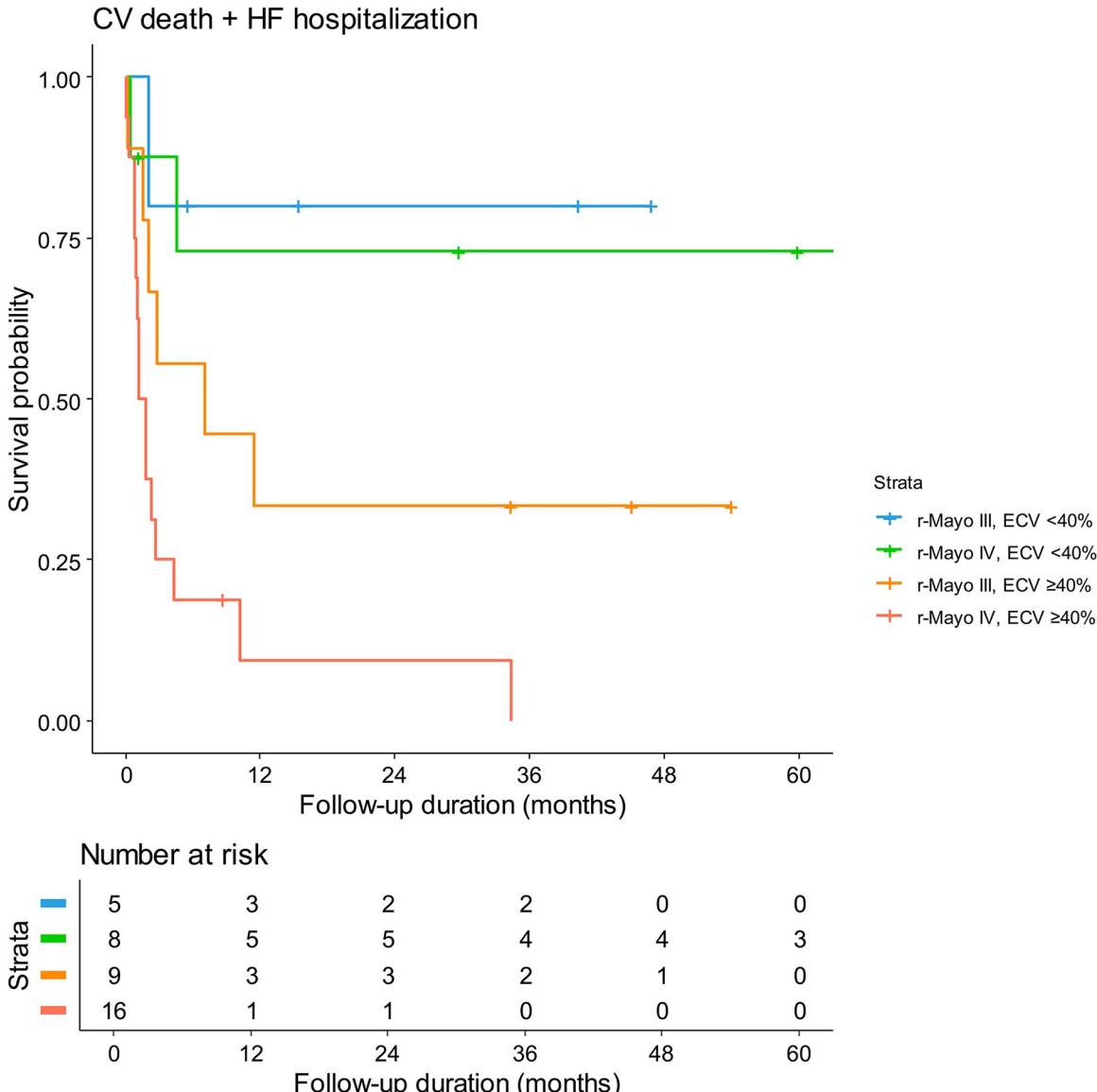

**Fig 5. Event-free survival of patients with AL-CA according to the combination of revised Mayo stage and automated ECV measurement.** Clinical outcomes were compared between the subgroups divided by a combination of revised Mayo staging and automated ECV. Abbreviations as in Fig 1.

TTR-CA were dichotomized according to the median value of automated ECV (40%), and the prognosis was compared between those with ECV <40% and those with ECV ≥40% (**S4 Fig**). Although we could not obtain statistically significant results due to the small number of patients, it was observed that the patients with TTR-CA and higher ECV levels (automated ECV ≥40%) tended to have higher risk of composite outcomes (cardiovascular death and heart failure hospitalization) compared to those with lower ECV levels (<40%) (Log-rank p = 0.218; unadjusted HR 3.629, 95% CI 0.405–32.556, p = 0.250).

## Discussion

In this study, we analyzed the performance of the AI-automated segmentation T1 mapping in the diagnosis and prognostication of CA. The results demonstrated that the automated native

**Table 3. Incremental prognostic value of automated ECV measurement over revised Mayo staging.**

| | Value | 95% CI | p-value |
|---|---|---|---|
| IDI (combination of revised Mayo stage and ECV vs. revised Mayo stage only) | 0.279 | 0.025–0.499 | 0.013 |
| NRI (combination of revised Mayo stage and ECV vs. revised Mayo stage only) | 0.138 | 0.000–0.597 | 0.007 |
| AUC of the combination of revised Mayo stage and ECV | 0.860 | 0.740–0.979 | 0.006 |
| AUC of revised Mayo stage | 0.588 | 0.427–0.748 | |

Abbreviations: AUC, area under the curve; IDI, integrated discrimination improvement; NRI, net reclassification improvement.

T1 and ECV were effective in differentiating CA from other etiologies of LVH. Furthermore, in patients with AL-CA, the automated ECV showed independent prognostic value beyond the revised Mayo stage and could provide better risk stratification when incorporated into the current staging system. Our findings suggest the usefulness of automated T1 mapping for differential diagnosis of LVH, as well as for prognostication in AL-CA (**S5 Fig**).

## Etiologies of LVH and the importance of differential diagnosis

LVH is a common cardiac condition, but various etiologies should be considered because of the considerable differences in the pathophysiology, treatment, and prognosis across the possible etiologies of LVH [1,3]. If LVH is a result from increased afterload, i.e., HHD, management is mainly focused on the alleviation of blood pressure, and the prognosis is benign. However, in patients with HCM, the assessment of sudden cardiac death risk and its prevention is crucial in the treatment strategy, with special considerations required for its complications, such as arrhythmia, heart failure, or LV outflow tract obstruction [8,9]. AL-CA requires a complex approach, given that AL-CA is a hematologic malignancy with excessive production of light-chain immunoglobulins [2]. Its management includes cytotoxic chemotherapy and stem cell transplantation, along with the management of cardiovascular complications, but the prognosis is poor: patients with AL-CA have a median survival of 24 months from the initial diagnosis [5]. As a subtype of CA, TTR-CA shares common morphological features with AL-CA; however, its pathophysiology is unique. The accumulation of TTR, due to genetic or unknown causes, is the main target for specialized treatment (i.e., TTR tetramer stabilizers) [10]. Another unique etiology of diffuse LVH should be noted: FD is a rare X-linked lysosomal storage disorder, caused by mutations in *GLA*, responsible for coding the lysosomal enzyme α-galactosidase A [11]. The absent or insufficient α-galactosidase A activity leads to the accumulation of globotriaosylceramide and related globotriaosylsphingosine in affected tissues. This accumulation in myocardium results in myocardial fibrosis, requiring specific therapies, primarily enzyme replacement therapy.

## Differential diagnosis of LVH etiologies using CMR

The morphological similarities of different etiologies of LVH and the lack of pathognomonic/specific findings makes its differential diagnosis using echocardiography challenging [3]. Therefore, an arbitrary diagnosis based on echocardiography needs to be investigated further using non-invasive imaging modalities, such as CMR. In particular, HCM typically shows multifocal discrete LGE, CA shows diffuse LGE typically at subendocardial layer, FD shows diffuse LGE across entire myocardium, and HHD shows focal fuzzy LGE at the right ventricular insertion points of LV myocardium [12]. However, these patterns can overlap between etiologies, and may not even be apparent in the early stages of the disease. Given these limitations and

ambiguity in the assessment of LGE patterns for the differential diagnosis of LVH etiologies, several studies have been conducted to analyze the efficacy of quantitative assessment using native T1 and ECV [6,15]. Native T1 and ECV typically reflect the degree and amount of myocardial fibrosis, not only in the advanced stage (so called "replacement fibrosis") but also in the early stage of diffuse myocardial fibrosis [16]. The accumulation of robust evidence has led to steady evolution of the use of native T1 and ECV in myocardial disease, showing different patterns of native T1 and ECV between the etiologies of LVH. According to a study by Baggiano et al., native T1 enabled diagnosis of CA, with a cutoff native T1 of <1036 msec resulting in a 98% negative predictive value and native T1 value of >1164 msec resulting in a 98% positive predictive value [17]. Hinojar et al reported that patients with HCM show significantly higher native T1 and ECV than those with HHD [18]. FD is also known to have significantly lower native T1 values than HHD and HCM [19,20]. Our findings are consistent with previous studies, showing that CA results in the highest native T1 and ECV, followed by HCM and HHD, and with FD showing the lowest values.

Although both native T1 and ECV can be useful in differential diagnosis of LVH etiologies, it should be noted that native T1 can be affected by MR vendors, the intensity of magnetic field, and the amount and concentration of contrast agent. On the other hand, ECV represents a physiological parameter and is derived from the ratio of T1 signal values, therefore could be more reproducible between different field strengths, vendors, and acquisition techniques than both native and post-contrast T1 [15]. ECV measures also exhibit better agreement with histological measures of the collagen volume fraction than isolated post-contrast T1. Given the better performance of ECV (AUC 0.946) in differentiating CA from other etiologies of LVH than that of native T1 (AUC 0.899), our findings support that ECV measurements could be more relevant than native T1 values for diagnosis of CA.

## Deep learning algorithm for the measurement of native T1 and ECV

A previous study of 95 participants, including 12 patients with HCM, 12 with CA, and 12 with FD, conducted using our DL model demonstrated excellent correlation and agreement between the automated native T1 and ECV and the reference values on a per-patient basis (native T1: r = 0.967, bias 9.5 msec; ECV: r = 0.987, bias 0.7%) [13]. The excellent accuracy of our DL algorithm, which was further confirmed in the present study, indicates that the use of DL algorithm on the T1 mapping images can considerably reduce the time required for image interpretation [13]. In addition, considering that the differential diagnosis of LVH requires a series of non-invasive and invasive tests, the accuracy of T1 mapping parameters in the detection of CA can improve the diagnostic process in clinical practice. In particular, the differential diagnosis process includes myocardial biopsy for confirming amyloid infiltration in myocardium with specific immunohistochemistry staining for determination of CA subtypes. Genetic tests are required for the confirmation of HCM and TTR-CA, a process that leads to some delays until the final diagnosis. Thus, accurate interpretation of CMR images using T1 mapping can focus down to the possible diagnosis and facilitate the selection of appropriate diagnostic tests.

## Prediction of prognosis in AL-CA using T1 mapping parameters

In the present study, we assessed the prognostic value of ECV, because in addition to ECV having theoretical advantages over native T1, the AUC of ECV in the present study was higher than that of native T1, not only for differentiating CA from other LVH etiologies but also for predicting its prognosis. Additionally, given the small number of events in patients with HHD and HCM, and the small number of patients with TTR-CA and FD, we focused our analysis on patients with AL-CA.

Several clinical tools have been used to assess the prognosis of patients with AL-CA. The revised Mayo staging system suggested in 2012, which is the most widely used clinical staging system for AL-CA, incorporates multiparametric biomarkers, such as the levels of NT-proBNP, cardiac troponin, and the dFLC [21]. However, it could be argued that the incorporation of myocardial status into the system would be more representative of the disease stage, because the most common cause of death in patients with AL-CA is cardiovascular death. Indeed, the incorporation of LGE on CMR image into the staging system provided promising results [22]. However, it should be noted that the determination of LGE in AL-CA is often obscure, due to the fuzzy involvement of amyloid. Instead, the T1 mapping could be promising, given its more feasible quantitation and direct reflection of the burden of amyloid infiltration in myocardium [23]. Moreover, patients with revised Mayo stage III and IV in the present study who had elevated ECV (indicative of advanced myocardial infiltration) demonstrated higher risks of mortality than those with lower ECV. Furthermore, ECV had a more significant predictive value than the components of the revised Mayo staging system, such as the levels of NT-proBNP, cardiac troponin, and dFLC. These findings suggest that direct assessment of myocardial fibrosis using T1 mapping techniques could be more suitable for the prediction of mortality risk than indirect measurement of amyloid infiltration using serum biomarkers. Additionally, given that this is the first study to demonstrate the relevance of DL model not only for the diagnosis of CA, but also for prognostication in patients with AL-CA, our findings will facilitate the clinical application of DL models in CMR imaging.

## Limitations

The present study has certain limitations. First, this was a single-center retrospective study with a limited sample size. Second, the prognostic value of T1 mapping parameters was assessed only among the patients with AL-CA, but not for those with other etiologies. We observed a trend toward worse prognosis in patients with TTR-CA and higher ECV (automated ECV $\geq 40\%$) compared to those with lower ECV; however, the number of TTR-CA cases in this study was limited to 11, necessitating future studies for statistically meaningful analysis. Patients with HCM and HHD did not experience sufficient number of events, and the number of patients with FD was too small for further analysis. Third, while the qualitative assessment of LGE, including its distribution and pattern, proved useful for differentiating LVH etiologies in our study population, it was not employed for prognostication due to the typically similar LGE patterns observed in patients with AL-CA. Additionally, we could not provide a quantitative assessment of LGE in this study. Instead, we focused on the clinical usefulness of the automated T1 mapping parameters by DL algorithm. Nonetheless, we acknowledge that future studies on the radiomics features of LGE in the automated CMR interpretation are warranted. Fourth, our study focused on the clinical utilization of automated measurements of T1 mapping parameters, rather than the automatization of the entire clinical process. Finally, due to the absence of a testing or validation cohort in this study, our findings should be interpreted with caution and warrant prospective validation in terms of improving the diagnostic process and prognosis.

## Conclusion

This study demonstrated that native T1 and ECV measurements obtained through an AI-automated segmentation T1 mapping on CMR images could differentiate CA from other etiologies of LVH. Furthermore, this study showed that automated ECV has significant prognostic value in patients with AL-CA, suggesting its usefulness in clinical practice.

## Supporting information

**S1 File. Detailed methods.**
(PDF)

**S1 Fig. Schematic figure of the native T1 and ECV measurements derived from AI-automated segmentation T1 mapping.**
(PDF)

**S2 Fig. Accuracy of automated native T1 and ECV measurements.**
(PDF)

**S3 Fig. ROC curves for the prediction of clinical outcomes of AL-CA using the automated native T1 and ECV measurements.**
(PDF)

**S4 Fig. Event-free survival of patients with TTR-CA according to the automated ECV.**
(PDF)

**S5 Fig. Graphical abstract: Automated ECV measurement for diagnosis and prognostication in AL-CA.** Abbreviations: ECV, extracellular volume fraction; AL-CA, AL cardiac amyloidosis; CV, cardiovascular; HF, heart failure; r-Mayo, revised Mayo staging.
(TIF)

**S1 Table. Univariable predictors of the composite outcome among patients with AL-CA.**
(PDF)

## Author Contributions

**Conceptualization:** In-Chang Hwang, Eun Ju Chun, Pan Ki Kim, Yeonyee E. Yoon, Goo-Yeong Cho.

**Data curation:** In-Chang Hwang, Eun Ju Chun, Jiesuck Park, Hong-Mi Choi, Yeonyee E. Yoon, Goo-Yeong Cho.

**Formal analysis:** In-Chang Hwang, Eun Ju Chun, Jiesuck Park, Hong-Mi Choi, Yeonyee E. Yoon.

**Funding acquisition:** In-Chang Hwang, Eun Ju Chun, Myeongju Kim.

**Investigation:** In-Chang Hwang, Eun Ju Chun, Jiesuck Park, Hong-Mi Choi, Yeonyee E. Yoon, Goo-Yeong Cho.

**Methodology:** In-Chang Hwang, Eun Ju Chun, Myeongju Kim, Jiesuck Park, Hong-Mi Choi, Yeonyee E. Yoon, Goo-Yeong Cho.

**Project administration:** In-Chang Hwang, Eun Ju Chun, Myeongju Kim, Byoung Wook Choi.

**Resources:** In-Chang Hwang, Eun Ju Chun, Myeongju Kim, Jiesuck Park, Hong-Mi Choi, Yeonyee E. Yoon, Goo-Yeong Cho.

**Software:** In-Chang Hwang, Eun Ju Chun, Pan Ki Kim, Hong-Mi Choi, Goo-Yeong Cho, Byoung Wook Choi.

**Supervision:** In-Chang Hwang, Eun Ju Chun, Goo-Yeong Cho, Byoung Wook Choi.

**Validation:** In-Chang Hwang, Eun Ju Chun.

**Visualization:** In-Chang Hwang, Eun Ju Chun, Jiesuck Park.

**Writing – original draft:** In-Chang Hwang, Eun Ju Chun, Pan Ki Kim, Myeongju Kim, Jiesuck Park, Hong-Mi Choi, Yeonyee E. Yoon, Goo-Yeong Cho, Byoung Wook Choi.

**Writing – review & editing:** In-Chang Hwang, Eun Ju Chun, Pan Ki Kim, Myeongju Kim, Jiesuck Park, Hong-Mi Choi, Yeonyee E. Yoon, Goo-Yeong Cho, Byoung Wook Choi.

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
