## [Decision Letter · Decision Letter 0]

6 Nov 2024

PONE-D-24-22784Automated Extracellular Volume Fraction Measurement for Diagnosis and Prognostication in Patients with Light-Chain Cardiac AmyloidosisPLOS ONE

Dear Dr. Hwang,

Thank you for submitting your manuscript to PLOS ONE and apologise for the delay in revising it. After careful consideration, we feel that it has merit but does not fully meet PLOS ONE’s publication criteria as it currently stands. Therefore, we invite you to submit a revised version of the manuscript that addresses the points raised during the review process.

We look forward to receiving your revised manuscript.

Kind regards,

Giuseppe Limongelli

Academic Editor

PLOS ONE

**Journal Requirements:**

This work was supported by the Medical AI Clinic Program through the National IT Industry Promotion Agency (NIPA) funded by the Ministry of Science and ICT (MSIT) of the Republic of Korea, and by research grants from Seoul National University Bundang Hospital (Grant No. 02-2017-0040 and 06-2020-0130).

Pan Ki Kim and Byoung Wook Choi are founders of Phantomics, Inc. (Seoul, Korea), which supports the software used in this study; however, they did not participate in the data analysis. The other authors declare no conflicts of interest.

We note that one or more of the authors are employed by a commercial company: Phantomics, Inc. (Seoul, Korea) 

**Additional Editor Comments **

Dear author

I apologise for the delay in the revision process. please find attached reviewers' comments below.

Reviewers' comments:

Reviewer's Responses to Questions

**Comments to the Author**

1. Is the manuscript technically sound, and do the data support the conclusions?

Reviewer #1: Yes

Reviewer #2: Yes

2. Has the statistical analysis been performed appropriately and rigorously? 

Reviewer #1: Yes

Reviewer #2: Yes

3. Have the authors made all data underlying the findings in their manuscript fully available?

Reviewer #1: Yes

Reviewer #2: Yes

4. Is the manuscript presented in an intelligible fashion and written in standard English?

Reviewer #1: Yes

Reviewer #2: Yes

5. Review Comments to the Author

**Reviewer #1:** the authors present an interesting study on the ability of AI automated segmentation T1 mapping to detect cardiac amyloidosis (CA) in patients with LVH. Also, they show that automated ECV measurements in patients with AL-CA improve prognostication of the disease.

It would be useful to clarify few points:

- were the patients with suspicion of LVH consecutive patients referred for MRI due to LVH ? if not, why?

- why was prognostic value of ECV calculated only for AL patients and not for ATTR as well?

**Reviewer #2: **The authors present a study on the use of a artificial intelligence (AI) -based T1 mapping in 300 patients diagnosed with left ventricular hypertrophy (LVH).

CMR AI-based T1-mapping was able to detect cardiac amyloidosis (CA) and provided prognostic information in those with AL-CA.

The potential use of AI based T1-mapping and automated extracellular volume (ECV) calculation in diagnosis and prognostication of LVH is clinically relevant.

While I understand quantitative assessment of LGE was not available to the authors, it would be useful to include how many patients presented with LGE and the distribution in the myocardium. Also,, CMR-derived LVEF and LVEDV measures should be included in Table 1.

Finally it should be highlighted in the limitations that there is no testing/validation cohort and the results might not be generalisable.

6. PLOS authors have the option to publish the peer review history of their article (what does this mean?). If published, this will include your full peer review and any attached files.

Reviewer #1: No

Reviewer #2: No

---

## [Author Response · Author response to Decision Letter 0]

1 Dec 2024

*** Please refer to the attached "Response Letter".

Point-by-point Responses to Reviewer #1 Comments to Author:

General comment

The authors present an interesting study on the ability of AI automated segmentation T1 mapping to detect cardiac amyloidosis (CA) in patients with LVH. Also, they show that automated ECV measurements in patients with AL-CA improve prognostication of the disease.

Response to the general comment

We thank the reviewer’s considerate review and comments. We believe that the revision processes have further improved the strengths of our study. We tried our best to clarify the raised comments, and if the issues were not able to be addressed within the present study, we honestly acknowledged the limitations of our study, in order to provide appropriate background information for the readers. We just hope that the revised version with responses to the reviewer can better comply with the publication standards of PLOS ONE.

Comment #1

It would be useful to clarify few points: were the patients with suspicion of LVH consecutive patients referred for MRI due to LVH ? If not, why? 

Response to comment #1

We sincerely thank the reviewer for this valuable comment. In the present study, we identified consecutive patients who underwent cardiac MRI for differential diagnosis of the etiology, between 2011 and 2023. In the revised manuscript, we indicated that the study population was retrospectively identified on a consecutive manner. 

Changes in accordance with comment #1

• Abstract (page 2): A total of 300 consecutive patients who underwent CMR for differential diagnosis of LVH were analyzed.

• Methods – Study design and cohort (page 4): We retrospectively identified 300 consecutive patients (47 with HHD, 198 with HCM, 50 with CA [39 with AL-CA, and 11 with TTR-CA], and 5 with Fabry disease [FD]) (Fig 1) who underwent CMR for differential diagnosis of the etiology, based on the detection of LVH on echocardiography, between 2011 and 2023.

Comment #2

Why was prognostic value of ECV calculated only for AL patients and not for ATTR as well?

Response to comment #2

Thank you for your valuable feedback. Among the total 300 consecutive patients who underwent CMR for differential diagnosis of LVH, we have found 50 patients with cardiac amyloidosis, consisting of 39 patients with AL-CA and 11 patients with TTR-CA. Those 11 patients with TTR-CA were further divided according to the subtype of TTR: 3 patients with hereditary TTR-CA (genetic; hATTR) and 8 patients with wild-type TTR-CA (senile; wtATTR). Given the number of patients in each ATTR subgroup (hATTR and wtATTR), we performed survival analysis among the total 11 patients with TTR-CA according to the ECV. Although we could not obtain statistically significant results due to the small number of patients, it was observed that the patients with TTR-CA and higher ECV levels (automated ECV ≥40%) tended to have higher risk of composite outcomes (cardiovascular death and heart failure hospitalization) compared to those with lower ECV levels (<40%).

 In the revised manuscript, we added these findings as Supplementary results and provided interpretations in the Limitation section, as below:

Changes in accordance with comment #2

• Results (page 16): Prognostic value of ECV in patients with TTR. The prognostic value of the automated ECV was further assessed in patients with TTR-CA. Among the 11 patients with TTR-CA included in the present study (3 with hereditary TTR-CA [genetic; hATTR] and 8 with wild-type TTR-CA [senile; wtATTR]), 1 patient died of cardiovascular cause and 6 patients experienced hospitalization for heart failure. The AUC values for the study outcome using the automated native T1 and ECV were 0.600 (95% CI 0.226–0.974; p=0.584) and 0.700 (95% CI 0.376–1.000; p=0.273), respectively. These patients with TTR-CA were dichotomized according to the median value of automated ECV (40%), and the prognosis was compared between those with ECV <40% and those with ECV ≥40% (S4 Fig.). Although we could not obtain statistically significant results due to the small number of patients, it was observed that the patients with TTR-CA and higher ECV levels (automated ECV ≥40%) tended to have higher risk of composite outcomes (cardiovascular death and heart failure hospitalization) compared to those with lower ECV levels (<40%) (Log-rank p=0.218; unadjusted HR 3.629, 95% CI 0.405–32.556, p=0.250).

• S4 Fig. Event-free survival of patients with TTR-CA according to the automated ECV

• Limitation (page 21): Second, the prognostic value of T1 mapping parameters was assessed only among the patients with AL-CA, but not for those with other etiologies. We observed a trend toward worse prognosis in patients with TTR-CA and higher ECV (automated ECV ≥40%) compared to those with lower ECV; however, the number of TTR-CA cases in this study was limited to 11, necessitating future studies for statistically meaningful analysis. Patients with HCM and HHD did not experience sufficient number of events, and the number of patients with FD was too small for further analysis.

Point-by-point Responses to Reviewer #2 Comments to Author:

General comment

The authors present a study on the use of an artificial intelligence (AI) -based T1 mapping in 300 patients diagnosed with left ventricular hypertrophy (LVH). CMR AI-based T1-mapping was able to detect cardiac amyloidosis (CA) and provided prognostic information in those with AL-CA.

The potential use of AI based T1-mapping and automated extracellular volume (ECV) calculation in diagnosis and prognostication of LVH is clinically relevant.

Response to the general comment

We appreciate the reviewer’s thoughtful comments and feedback. We believe that the revision process has helped enhance the strengths of our study. We have made every effort to address the comments raised, and where certain issues could not be fully resolved within the scope of this study, we have acknowledged these limitations to provide readers with a clear context. We hope that the revised version now aligns more closely with the publication standards of PLOS ONE.

Comment #1

While I understand quantitative assessment of LGE was not available to the authors, it would be useful to include how many patients presented with LGE and the distribution in the myocardium. 

Comment #2

Also, CMR-derived LVEF and LVEDV measures should be included in Table 1.

Response to comments #1 and #2

We sincerely thank the reviewer for this valuable comment. According to the reviewer’s advice, we have added the assessment of LGE, as well as LV volumetric measures. The LV-EDV was the smallest in patients with AL-CA (74.0±24.9 mL), followed by those with TTR-CA (81.2±24.4 mL), and was the largest in patients with Fabry disease (107.7±57.8 mL). LV-EF was similar across the subgroups of HHD, HCM, and Fabry disease, but was significantly lower in patients with AL-CA (59.0±11.0%). 

 It was noted that LGE was observed in most of the patients with HCM (97.0%), AL-CA (97.4%), TTR-CA (100.0%), and Fabry disease (80.0%), whereas 40% of patients with HHD had LGE, which mainly showed focally-distributed patchy LGE. Typically, patients with AL-CA and TTR-CA showed diffuse distribution of LGE with subendocardial ring-type enhancement, and patients with HCM showed focal or multifocal distribution of LGE with patchy segmental enhancement. It was noted that the extent and distribution pattern of LGE were not associated with study outcomes, because of the typically similar LGE patterns observed in patients with AL-CA (35 patients [89.7%] of AL-CA had diffuse LGE with subendocardial ring-type distribution).

 In the revised manuscript, we added descriptions regarding the LV volumetric measures and LGE assessment findings in the Methods section, Results section, Limitations section, and Table 1, as below. 

Changes in accordance with comments #1 and #2

• Methods (pages 6 – 7): LGE extent was categorized into focal (localized distribution), multifocal (distribution across several locations), and diffuse (involving at least three contiguous myocardial segments). Additionally, the distribution patterns were classified into six distinct types and analyzed as follows; patchy segmental, transmural, mid-layer, RV insertion point, subendocardial ring, and nonspecific.

• Results (page 9): The mean LVMI was 118.7±33.7 g/m2 in patients with HHD, 134.9±32.7 g/m2 in those with HCM, 134.0±36.1 g/m2 in those with AL-CA, 143.0±54.0 g/m2 in those with TTR-CA, and 162.2±80.2 g/m2 in those with FD. LV-EDV measured by CMR was smallest in patients with AL-CA (74.0±24.9 mL), followed by those with TTR-CA (81.2±24.4 mL), and largest in patients with Fabry disease (107.7±57.8 mL). LV-EF measured by CMR was comparable among patients with HHD, HCM, and Fabry disease but was significantly lower in those with AL-CA (59.0±11.0%). LGE was observed in the majority of patients with HCM (97.0%), AL-CA (97.4%), TTR-CA (100.0%), and Fabry disease (80.0%). In contrast, only 40% of patients with HHD showed LGE, which primarily presented as focal, patchy enhancement. Patients with AL-CA and TTR-CA typically exhibited diffuse LGE with a subendocardial ring-like pattern, while those with HCM demonstrated focal or multifocal LGE with patchy, segmental enhancement. Patients with CA had the highest native T1 values (1444.6±83.4 msec in AL-CA, and 1418.3±72.8 msec in TTR-CA), followed by patients with HCM (1319.4±56.9 msec), and those with HHD (1291.4±38.9 msec), and patients with FD (1143.9±80.3 msec) (Fig 3A).

• Results (page 13): The extent and distribution pattern of LGE were not associated with the study outcomes due to the predominantly similar LGE patterns observed in patients with AL-CA: specifically, 35 patients (89.7%) with AL-CA exhibited diffuse LGE with a characteristic subendocardial ring-type distribution (Table 1 and S1 Table).

• Limitations (page 21): Third, while the qualitative assessment of LGE, including its distribution and pattern, proved useful for differentiating LVH etiologies in our study population, it was not employed for prognostication due to the typically similar LGE patterns observed in patients with AL-CA. Additionally, we could not provide a quantitative assessment of LGE in this study. Instead, we focused on the clinical usefulness of the automated T1 mapping parameters by DL algorithm. Nonetheless, we acknowledge that future studies on the radiomics features of LGE in the automated CMR interpretation are warranted.

• Revised Table 1. Baseline characteristics

• Revised S1 Table. Univariable predictors of the composite outcome among patients with AL-CA

Comment #3

Finally, it should be highlighted in the limitations that there is no testing/validation cohort and the results might not be generalisable.

Response to comment #3

We agree with the reviewer’s valuable comment. According to the reviewer’s advice, we added in the limitation that, because of the lack of testing/validation cohorts, our findings should be interpreted with caution and might not be generalizable in other study populations or different clinical settings.

Changes in accordance with comment #3

• Limitations (page 21): Finally, due to the absence of a testing or validation cohort in this study, our findings should be interpreted with caution and warrant prospective validation in terms of improving the diagnostic process and prognosis.

Point-by-point Responses to the Journal Requirements:

Item #1

Response to Item #1

In the revised manuscript, we followed the PLOS ONE’s style requirements.

Item #2

Please note that PLOS ONE has specific guidelines on code sharing for submissions in which author-generated code underpins the findings in the manuscript. In these cases, all author-generated code must be made available without restrictions upon publication of the work. Please review our guidelines at https://journals.plos.org/plosone/s/materials-and-software-sharing#loc-sharing-code and ensure that your code is shared in a way that follows best practice and facilitates reproducibility and reuse.

Response to Item #2

According to the PLOS ONE’s guidelines on code sharing, we provided the codes for statistical analyses in the revised manuscript files (https://github.com/Inchang-Hwang/CMR-ECV-for-AL-CA). Regarding the automated measurements of native T1 and ECV on cardiac MR images, we believe that the deep learning algorithm for automatically measuring native T1 and ECV in cardiac MRI images is not directly relevant to the content of this study, as it is incorporated into an analysis program whose accuracy has already been validated in multiple prior studies (Ref: Korean J Radiol. 2022;23(12):1251-9 / Korean J Radiol. 2023;24(5):395-405). Additionally, as this analysis program, which includes the deep learning algorithm, is commercialized by Phantomics Inc., we ask for your understanding that specific details cannot be disclosed due to company confidentiality.

Below is the revised Data Availability Statement.

• Data availability statement: Data used in this study cannot be made publicly available because of the strict ethical restrictions set by the IRB of Seoul National University Bundang Hospital (https://e-irb.snubh.org). Please contact the corresponding authors (inchang.hwang@gmail.com or humandr@snubh.org) or the ethics board at SNUBH (snubhirb@gmail.com) for further inquiries regarding data availability within the scope permitted by the IRB. The code for statistical analysis utilized in the present study was released (https://github.com/Inchang-Hwang/CMR-ECV-for-AL-CA).

Item #3

Thank you for stating in your Funding Statement: 

This work was supported by the Medical AI Clinic Program through the National IT Industry Promotion Agency (NIPA) funded by the Ministry of Science and ICT (MSIT) of the Republic of Korea, and by research grants from Seoul National University Bundang Hospital (Grant No. 02-2017-0040 and 06-2020-0130).

Response to Item #3

We confirm that there was no additional external funding received for this study. According to the journal’s requirements, we amended Funding Statement, Competing Interests Statement, and Author Contributions Statement, as below: 

• Funding: This work was supported by the Medical AI Clinic Program through the National IT Industry Promotion Agency (NIPA), funded by the Ministry of Science and ICT (MSIT) of the Republic of Korea, and by research grants from Seoul National University Bundang Hospital (Grant Nos. 02-2017-0040 and 06-2020-0130). Pan Ki Kim and Byoung Wook Choi are founders of Phantomics, Inc. (Seoul, Korea), which provided support for the software used in this study. Phantomics, Inc. also funded salaries for PKK and BWC but had no role in the study design, data collection, analysis, publication decisions, or manuscript preparation. The specific contributions of each author are detailed in the ‘author contributions’ section. No additional external funding was received for this study.

• Competing Interests: Pan Ki Kim and Byoung Wook Choi are founders of Phantomics, Inc. (Seoul, Korea), which provided support for the software used in this study. Phantomics, Inc. also funded salaries for PKK and 

---

## [Editor Report · Decision Letter 1]

6 Jan 2025

Automated extracellular volume fraction measurement for diagnosis and prognostication in patients with light-chain cardiac amyloidosis

PONE-D-24-22784R1

Dear Dr. In-Chang Hwang

We’re pleased to inform you that your manuscript has been judged scientifically suitable for publication and will be formally accepted for publication once it meets all outstanding technical requirements.

Kind regards,

Giuseppe Limongelli

Academic Editor

PLOS ONE

---

## [Editor Report · Acceptance letter]

9 Jan 2025

PONE-D-24-22784R1 

PLOS ONE

Dear Dr. Hwang, 

I'm pleased to inform you that your manuscript has been deemed suitable for publication in PLOS ONE. Congratulations! Your manuscript is now being handed over to our production team.

Kind regards, 

on behalf of

Dr. Giuseppe Limongelli 

Academic Editor

PLOS ONE